# Most Up-to-Date Analysis of Epidemiological Data on the Screening Guidelines and Incidence of Retinopathy of Prematurity in Europe—A Literature Review

**DOI:** 10.3390/jcm12113650

**Published:** 2023-05-24

**Authors:** Monika Modrzejewska, Wiktoria Bosy-Gąsior

**Affiliations:** 12nd Department of Ophthalmology, Pomeranian Medical University in Szczecin, Powstańców Wielkopolskich 72, 70-111 Szczecin, Poland; 2Scientific Association of Students 2nd Department of Ophthalmology, Pomeranian Medical University in Szczecin, Powstańców Wielkopolskich 72, 70-111 Szczecin, Poland; wiktoriazbosy@gmail.com

**Keywords:** retinopathy of prematurity (ROP) in Europe, epidemiology of ROP, screening for ROP, percentage of accurate diagnoses of ROP, percentage of premature infants treated for ROP

## Abstract

Introduction: Global predictions indicate that the incidence of retinopathy of prematurity (ROP) is increasing, but the lack of current epidemiological data on the occurrence of ROP in Europe prompted the authors to update these data. Methods: European studies related to the presence of ROP were analyzed, and the reason for the differences in the percentage of ROP and different screening criteria were checked. Results: The study presents results from individual and multiple centers. Current ROP incidence data varies from a low of 9.3% in Switzerland to the highest values of 64.1% in Portugal and 39.5% in Norway. The national screening criteria are used in the Netherlands, Germany, Norway, Poland, Portugal, Switzerland, and Sweden. Uniform criteria—Royal College of Paediatrics and Child Health guidelines are used in England and Greece. American Academy of Pediatrics screening guidelines are used in France and Italy. Conclusions: The epidemiology of ROP in European countries varies significantly. The increase in the diagnosis and treatment rate of ROP in recent years correlates with the narrowing of diagnostic criteria in newly issued guidelines (which include the WINROP and G-ROP algorithms), a higher number of less developed preterm infants, and a lower percentage of live births.

## 1. Introduction

Retinopathy of prematurity (ROP) is a retinal vasoproliferative disorder affecting prematurely born infants, detected in 21.6% of this group between 33–34 weeks of age [1]. One of the most well-known risk factors is the use of oxygen therapy in neonatal intensive care and the immaturity of preterm infants (i.e., low birth weight and early week of birth). Other, more overlooked risk factors include prenatal, perinatal, and maternal factors (i.e., methods of assisted reproduction, pregnancy-induced hypertension, thyroid diseases, gestational diabetes, chorioamnionitis and Preterm Premature Rupture of Membranes (PPROM)), and neonatal factors (mainly related to diseases of prematurity, i.e., neonatal sepsis (NS), respiratory distress syndrome (RDS), bronchopulmonary dysplasia (BPD), necrotizing enterocolitis (NEC), intraventricular hemorrhage (IVH), leukomalacia (L), and thrombocytopenia (T) [2]. Complications of this condition can pose a significant threat to the organ of vision leading to significant visual impairment and even blindness. It is estimated that today, 30,000–50,000 children worldwide have lost their sight as a consequence of the lack of favorable treatment results for severe ROP [3,4].

Observations by many researchers indicate that the incidence of ROP has been increasing over the years and tends to rise consistently. U.S. centers, during a 12-year follow-up, recorded an increase in the incidence of this disease by as much as 5.18% (2000–2012) [5]. An 8-year Dutch study (2009 vs. 2017) showed an increase in both the incidence of ROP by 6.4% and severe ROP by 2.3% [6]. Screening tests are performed in about 85% of countries worldwide, and 88% of those countries have defined screening criteria. These include countries in North, Central, and South America, Europe, Asia, and Oceania [7]. Screening criteria vary across European countries. In England and Greece, the recommendations of the Royal College of Paediatrics and Child Health [8] are used, while in France and Italy, the recommendations of the American Academy of Pediatrics are followed [9]. In other countries, internal guidelines, which are either uniform for all centers across the country (Poland, Portugal) or vary between the different centers (Switzerland, Sweden), are used.

To date, only one article (2019, by Kaur et al. [10]) was found in search engines, such as PubMed, Google Scholar, or UpToDate, which is a summarized publication of the literature review of various epidemiological studies of ROP incidence in Europe. In our paper, we presented an up-to-date review of the existing epidemiological literature not only on the incidence of ROP in European countries but also on the current state of the art of screening methods for ROP.

The available epidemiological data related to the occurrence of ROP on the European continent are scarce in the available literature, which we analyzed, such as search engines PubMed and Google Scholar/years 2018–2023. The lack of current epidemiological data has become a factor that triggers the desire to take a closer look at this still important issue, which in many poorly diagnosed cases contributes to the deterioration of vision and disorders in the eye.

Therefore, Internet databases related to the standardization of screening tests were searched with great interest, which differed in the guidelines used in the countries analyzed by us. The update on the occurrence of ROP in Europe, based on the analysis of pre-screening tests, is presented below.

## 2. Methods

This article is an epidemiological review of the incidence of ROP using literature from the last 5 years (2018–2023) from 11 European countries: England; France; Greece; Netherlands; Germany; Norway; Poland, Portugal; Switzerland; Sweden; and Italy. In our analysis, we focused on research conducted between 2010 and 2021. However, it was not possible to standardize the observation period in years for all countries due to the varied observation period of ROP occurrence by different authors.

The main analysis of our data was based on the percentage of ROP in screening studies conducted in various European countries as per the available reported data, and the percentage of ROP requiring treatment was recorded.

Risk factors for ROP development were not analyzed in this paper. It has been shown that screening guidelines in individual countries differed from one another and presented in Results.

So far, no generalized registry of ROP incidence has been introduced in most European countries. Many of these countries maintain individual databases, which are published at irregular intervals, adding to the challenge of standardizing results from specific periods.

## 3. Results

The results include diverse and heterogeneous groups of preterm infants. The data contains the number of preterm infants screened and those with an already established diagnosis of ROP, but there are no other demographic characteristics or parameters related to prematurity that fully define the characteristics of the groups of preterm infants studied. Epidemiological studies in Europe are conducted by individual bodies, as in France [11], Greece [12,13], Italy [14,15], Portugal [16,17,18], or multiple institutions, as in the Netherlands (80 centers) [6,19], Norway (Norwegian Neonatal Network, NNN) [20], Switzerland (SwissNeoNet registry) [21], or Sweden (SWEDROP register) [22,23,24]. Some articles contain the data of both individual and multiple centers. For example, in England, there are Bristol Eye Hospital (individual) [25], NHS operational data (161 medical units), the British Ophthalmic Surveillance Unit, and a national collaborative ROP special interest groups (327 groups) (multiple) [26,27]. In Germany, there are Bonn/Freiburg cohort ROP Registry [28], a single medical center in Hannover (individual) [29], ROP Registry, and the IQTIG database (multiple) [28]. In Poland, there are Adam Mickiewicz Medical University Hospital in Poznan, two large regions of Poland (individual) [30], and the GOCC database (multiple) [31]. Epidemiological data obtained from a review of available findings in the literature from European countries is presented in Table 1.

Differences between screening guidelines in individual countries are presented in Table 2. 

### 3.1. RetCam as an Important ROP Screening Device

In the literature reviewed by the authors, no diagnostic methods have been described so far that were used in the screening of ROP. It must be emphasized that the RetCam is a key tool in early screening diagnostics and follow-up examinations in the treatment of ROP. The 2004 study by Wu et al. [42] presented at the 30th Annual Meeting of the American Association for Pediatric Ophthalmology and Strabismus proved the high effectiveness of the RetCam device in the early diagnosis of ROP, as well as in reducing the overall number of required intermediate ophthalmological examinations in premature infants.

In 2010, an article published by Salcone et al. [43] described the RetCam as the most commonly used system for fundus photography due to its high-quality photographs and high reliability and accuracy in detecting referral-warranted ROP, particularly at later postmenstrual ages. In addition, it has been shown to be well-received by parents as well as cost-effective.

A study by Tejada-Palacios et al. [44], compared the method with the indirect binocular ophthalmoscope (the gold standard for ROP screening at the time of publication Tejada-Palacios et al.) with the RetCam device and showed the usefulness of the device in telemedicine and screening tests, as well as in the storage of retinal images, the ability to track the development of the disease, and to adjust the appropriate treatment [45].

Currently, new methods of digital diagnostics, such as Smartphone-Based Photography for Retinopathy of Prematurity, are becoming more and more popular. A study by Lin et al. [46], compared the RetCam device with the digital method of the C3 Funduscam device, showcasing the continuous imperfections of the smartphone invention with its limited ability to view the peripheral retina and properly identify the zone and stage of ROP, thus still working in favor of the RetCam device. In a study by Lekha et al. [47], however, it was recognized that the smartphone method is definitely future-proof. In terms of advantages, the low costs associated with the study, as well as the convenience of the method (required only smartphone + smartphone application + portable lens), were highlighted.

### 3.2. England

British studies conducted in various periods focused mainly on compiling information regarding the screening and treatment of ROP. The first of these, a study by Wong et al. [26], based on national cohort data using NHS operational data from the National Neonatal Research Database (NNRD), examined 161 (94%) medical units in England from 2009–2011. The total number of premature babies screened was 16,411, while 383 (3%) of them received ROP treatment. The article did not find information on the total number of children diagnosed with ROP. The number of screenings in the UK was reported, and the possibility of improving them for faster diagnosis and early treatment of ROP was discussed. The study by Tavassoli et al. [25], is a report from the Bristol Eye Hospital unit based on data from 2009–2015. The total number of children screened was 1152; 95 (8.2%) of them received treatment. An increase of 8.1% in children requiring treatment was observed (2009: 5.7%; 2015: 13.8%), which, similarly to the Dutch study by Trzcionkowska et al. [6], was linked to a higher number of surviving preterm babies, among other factors. The treatment of ROP was mainly laser therapy in 86% (82 out of 95), while an additional second diode laser treatment was required in 14% of children (13 out of 95), including only 5.3% (5 out of 95) with diode laser only, diode laser with anti-VEGF injection in 7.4% (7 out of 95), and diode laser with subsequent vitrectomy in 1.1% [25]. A nationwide population-based study of 327 groups across England by Adams et al. [27], was performed between December 2013 and December 2014 by the British Ophthalmic Surveillance Unit and a National Collaborative ROP Special Interest Group. The total number of premature infants included in the screening was 8112. Treatment for ROP was given to 4% of the (327 out of 8112) children with various stages of ROP. Severe A-ROP was confirmed in 8.26% (27 out of 327). The most common procedure performed in the analyzed group was laser photocoagulation: 89% (291 out of 327), while 8% (26 out of 327) of premature babies received an anti-VEGF preparation. The average parameters in the studied group of newborns were, respectively, 25 (GA) and 706 g (BW). The study also confirmed a 2.5-fold higher incidence of ROP treatment in the UK than previously estimated.

### 3.3. France

The results of the only French epidemiological study available on PubMed are presented by Chan et al. [11], They were collected from 2009–2015 and were derived from the medical database of the Neonatal Unit of the Bordeaux Hospital. The total number of premature babies screened during the period was 419; ROP was diagnosed in 27.7% of them (116 out of 419). Moreover, severe ROP was diagnosed in 9.5% of babies (11 out of 116) with a mean GA of 28 (±1.8) weeks and BW of 1034 g (±263).

### 3.4. Greece

The only body conducting long-term epidemiological studies of ROP is the Papageorgiou General Hospital of Thessaloniki. Two articles were found in the literature with data collected at different periods. The first study by Mataftsi et al. [13], was created using data from 2004–2015. There, screening tests were performed on 1178 premature infants. ROP was diagnosed in 19.7% of them (232 out of 1178). Severe stages of ROP were confirmed among 7.39% (232 out of 1178), while 2.5% of children were treated (30 out of 1178). The study by Moutzouri et al. [12], extended the previous study by another 4 years, thus forming a compilation of 16 years of data from 2004–2020. A total of 1643 premature infants were screened throughout the period. The study excluded 83 of them and qualified 1560. ROP was diagnosed in 18.5% of newborns (288 out of 1560), demonstrating an increase in the diagnosis of ROP compared to the previous study. Treatment was required in 3.14% of children (49 out of 1560). In the analyzed group of premature infants requiring therapy, the average GA was 26 (±2) weeks, and BW was 776.6 g (±217). These results suggest that the prevalence of ROP in this center in Greece decreased over the 4-year study extension period (19.4% vs. 18.5%), while the proportion of ROP requiring treatment increased (2.5% vs. 3.14%).

### 3.5. The Netherlands

Two articles were found describing the cases of premature infants diagnosed with ROP in the Netherlands. The first is a study by Trzcionkowska et al. [19], based on data from the neonatal intensive care units of 10 Dutch hospitals. This study encompasses seven years of observations on the development of ROP covering two separate periods, in which different guidelines for ROP diagnosis were adhered to (earlier period—NEDROP 1 January 2010–31 March 2013; later period—ETROP 1 April 2014–31 December 2016). The study was a comparison of these two periods to check the improvement of ROP diagnosis with the change of screening guidelines in a group of 11,295 premature infants, of which 196 were treated (1.7%): 1.1% (57 out of 5276) during the period when the NEDROP guidelines prevailed, and 2.3% (139 out of 6019) for the ETROP guidelines. Among all children, stage 3 (or higher) ROP was found in 49.1% of infants in 2010–2013 and in 57.6% in 2014–2016. The earlier period of the study included 57 premature infants, while the later period—a group of 139 children. The second study by Trzcionkowska et al. [6], compared data from 2009 with that from 2017 to observe ROP epidemiological data. As many as 80 Dutch hospitals were involved in the 2017 screening study, where 933 premature infants were submitted for statistical analysis. ROP was diagnosed in 28.3% of them (264 out of 933), and severe stage ROP was diagnosed in 4.4% of newborns (41 out of 933), and there was a lack of information about treatment. After comparative analyses of the results from 2009 (NEDROP-1 criteria used), in which the percentage of ROP was 21.9%, there was an increase in the diagnosis of ROP by as much as 6.4%. This phenomenon is explained by the narrowing of the diagnostic criteria in the NEDROP-2 guidelines (fewer infants at low risk of developing ROP), more premature preterm infants, as well as a lower percentage of live births in 2017.

### 3.6. Germany

The Larsen et al. [28], study features information from three separate databases: IQTIG (2010–2017), Bonn/Freiburg (2012–2016), and ROP-Registry (2011–2018). This study focuses on evaluating the development of ROP, including severe ROP, in a group of screened preterm infants born GA > 30. During the 2011–2018 period, 52,461 premature infants registered in the three databases mentioned above were screened. The IQTIG database recorded the following data: 33.1% of preterm infants with GA ≥ 30 (*n* = 17,347); 14.3% with GA = 30 (*n* = 7493); 10.2% with GA = 31 (*n* = 5363); and 8.6% with GA ≥ 32 (*n* = 4491). In addition, IQTIG showed the incidence of severe ROP requiring treatment in 2.9% of preterm infants (21), while the ROP Registry (a registry containing only preterm infants after treatment for severe ROP) showed that 1.42% of infants were diagnosed with severe ROP (281). Another Bonn/Freiburg registry contained 837 children screened: 38.5% (*n* = 321) with GA ≥ 30, including 13.9% (*n* = 116) with GA = 30, 11.9% (*n* = 100) with GA = 31, and 12.6% (*n* = 105) GA ≥ 32. 

Another study covering a wider range of years (2001–2017) is that of Akman et al. [29], from a single medical center in Hannover, which created the aforementioned retina.net ROP Registry database. It showed that ROP was diagnosed during the screening of 864 premature infants in the period of 2001–2017, and the proportion of premature infants requiring treatment was 7.5%. The same figure amounted to 4.1% in the 2006–2016 period. Of all the children treated, 89.2% (58 out of 65) underwent photocoagulation, and 10.8% (7 out of 65) received an anti-VEGF injection. The mean GA and BW values were, respectively, 25.7 (±1.8) weeks and 763 g (±235). Postmenstrual age at treatment was 38.2 weeks (±3.2).

### 3.7. Norway

The Grottenberg et al. [20], study is based on the Norwegian Neonatal Network (NNN) database from 2009–2017. This is a medical registry for newborns in neonatal units funded by the national government. The data includes 1156 live babies (1156 out of 1499; 77.1% mortality rate). During this period, ROP was diagnosed in 39.6% of them (458 out of 1156). Severe ROP developed in 13.1% (152 out of 1156), while 9.5% (110 out of 1156) of the preterm infants screened required treatment. A total of 65.1% (99 out of 152) of those diagnosed with severe ROP received treatment. A total of 63.7% of children (63 out of 99) received photocoagulation, 22.2% (22 out of 99) received anti-VEGF monotherapy, and 14.1% (14 out of 99) received combination therapy. The mean GA was 26.2 (±1.3) weeks, and the mean BW was 830 g (±203). The prevalence of ROP in Norway is exceptionally high. This may be due to both the high rate of premature infant mortality and the large amount of missing data. The country screens of all premature infants is <32 GA without narrowing down the criteria for ROP risk factors. The above factors may contribute to the overdiagnosis of ROP, leading to such a high prevalence of the disease.

### 3.8. Poland

The most recent report on the incidence of ROP in Poland is the study by Modrzejewska et al. [31], based on the 2012–2021 database from GOCC. In the given period, 97,214 premature infants were screened, and 15.6% of them were diagnosed with ROP (15,190 out of 97,214). In total, 31.8% (4837 out of 15,190) of premature infants were treated between 2012 and 2021. Laser therapy was used in 93.8% (4536 out of 4837), and anti-VEGF monotherapy was used in 2018–2021 (no data from previous years) in 6.2% of premature infants (301 out of 4837). In total, combination therapy (laser therapy and anty-VEGF) was performed in 5.8% of premature infants with ROP (279 out of 4837).

Another study by Chmielarz-Czarnocińska et al. [30], is based on the 2016–2019 database report from the Adam Mickiewicz Medical University Hospital in Poznan. The data included information from two large regions of Poland—Greater Poland and Lublin voivodeships. Screening tests were conducted on 1772 premature infants in these regions, while ROP was diagnosed in 459 of them, accounting for 25.9%. ROP treatment was administered to 23.5% (108 out of 459) of children diagnosed with ROP and to 6.1% after general screening (108 out of 1772). The mean gestational age and birth weight of the children treated in the aforementioned study was GA 26 (±2) weeks and BW 868 g (±236), respectively.

In the study by Modrzejewska et al. [31], the results from the GOCC database (including most centers in Poland) were unified with the data collected in the Chmielarz-Czarnocińska et al.’s study [30], (data from the areas covered by this study were not included in the GOCC reports). It was shown that the actual percentage of ROP occurrence in Poland in 2016–2019 was 15.5% (6353 out of 40,900) [31].

### 3.9. Portugal

The Almeida et al. study [16], which is the most recent epidemiological study of ROP from Portugal, is based on data from 2012–2020. In total, 475 premature infants were screened. A total of 23.8% were diagnosed with ROP (113 out of 475); severe ROP was found in 12.6% (60 out of 475), while 6.1% (29 out of 475) were treated. Studies from earlier years showed a significantly higher percentage of ROP diagnoses than the aforementioned study. The first is from Figueiredo et al. [17], which was conducted in 2008–2019 on a group of 239 premature babies undergoing screening. In this group of children, ROP was diagnosed in as many as 42.3% (101 out of 239), while severe ROP was diagnosed in 11.9% of preterm infants (12 out of 239). The mean BW and GA were, respectively, 1241.6 g (±310) and 29.8 (±3.4) weeks. Next is a study by Malheiro et al. [18], conducted between 2010 and 2016 at the Centro Hospitalar Universitário do Porto, on a group of 496 preterm infants undergoing screening, among whom ROP was diagnosed in 64.1% (275 out of 496), while only 5.8% of the preterm infants (26 out of 496) required treatment. A total of 92.3% of the newborns (24 out of 26) received diode laser treatment, while 7.7% (2 out of 26) underwent cryoablation. The mean GA was 31; BW was 1222.4 g.

### 3.10. Switzerland

The study by Gerull et al. [21] utilized the SwissNeoNet Registry, a national registry of preterm infants compiled from 2006–2015. The total number of preterm infants screened was 7817; however, after excluding mortality rates and missing data, 6472 children in this group were eligible for the study. ROP was diagnosed in 9.3% of preterm infants. Severe stage ROP was found among 1.8% of children (109 preterm infants), and 1.2% received treatment. It should be noted that Switzerland is likely to be one of the only countries with such a low rate of ROP. One possible reason for this is the provision of high-level medical care to premature infants, as well as the proper regulation and supply of oxygen during care in neonatal intensive care units. In addition, properly adjusted screening criteria (low risk of ROP between 29–32 GA) have been shown to reduce stress levels in preterm infants at low risk of developing ROP. In addition, in the years in which the analyses were carried out, a high mortality rate of extremely preterm infants was observed, which may also explain the lack of cases of premature infants with ROP and the lack of data on the incidence of ROP requiring treatment.

### 3.11. Sweden 

SWEDROP Register is a national registry for ROP screenings. The study created on the basis of the above-mentioned database (2008–2015) was a study by Holmström et al. [24], from 2018. During this period, 1157 premature infants GA ≥31 weeks were screened; then, after the introduction of new criteria, 5734 premature infants GA ≤ 30 weeks were screened. The % ROP results were only calculated within the group of premature infants GA ≤30 weeks, where it was found to be 31.9% (1829 out of 5734), with 5.7% (329 out of 5734) receiving treatment. When analyzing data from the above database, Holmström et al. [24], observed significant differences between the prevalence of ROP and treatment rates in seven Swedish regions. Another study based on the same database (from 2007–2018) was the study by Pivodic et al. [23], in 2020, in which 7609 premature infants were screened. ROP was diagnosed in 31.9% (2427 out of 7609), and 5.8% received treatment (442 out of 7609). The Holmström et al. [22], study consists of a database review of the incidence of ROP in Sweden from 2008 to 2017. A total of 8473 premature infants (7249 out of 8473 were <31 GA) were screened. ROP was diagnosed in 31.9% of them (2310 out of 7249), and 6.1% (440 out of 7249) were eligible for treatment. In the group of premature babies screened, 10.5% (46 out of 440) developed severe ROP. A total of 10.1% of the group was treated with anti-VEGF (44 out of 440), and combination therapy was applied to 17.3% (76 out of 440). The epidemiological data lack information on the size of the groups of premature infants in which laser-diode therapy was used. In addition to the three studies described, further publications from before 2016 were identified. All of them were based on one national database (still being updated) and focused on the screening and treatment of ROP in Sweden. New databases are created in order to establish the most advantageous screening method and ROP treatment guidelines in this country.

### 3.12. Italy

The two most recent epidemiological studies were published in 2021 from two separate regions in Italy—Tuscany and Genoa [14,15]. In the study by Dani et al. [15], from the Careggi University Hospital in Florence, between 2017 and 2020, screenings were performed on 178 premature infants. A total of 38% (67 out of 178) were diagnosed with ROP. Treatment was given to 16% by intravitreal delivery of anti-VEGF. In the study by Caruggi et al. [14], performed at the University of Genoa in 2015–2020, screenings were carried out within a group of 595 premature infants (of whom 120 were excluded); ultimately, 475 were included in the study. ROP was diagnosed in 25.1% of preterm infants (119 out of 475), which is a lower figure than that of the previous study. Treatment was required for 23.5% of children with ROP (28 out of 119). These studies provide evidence of a discrepancy in data obtained from different centers located geographically near each other, even in the same country, where the diagnosis of ROP could range from 25.1% up to 38%.

## 4. Discussion

On the basis of the analyzed epidemiological data, many differences in the percentage of ROP occurrence and in the percentage of ROP treated were observed between the countries described. The authors indicate that the differences in the reported numerical and percentage values of ROP occurrence result primarily from different modes of operation in screening tests in European countries adopted in the recommendations of a given country, whose task is to screen premature infants for newborns diagnosed with ROP, including clinically active ROP requiring treatment (Table 2).

Some countries use the same screening criteria. England and Greece are using the recommendations of the Royal College of Paediatrics and Child Health [8], and although English publications lack information on the diagnosis of % ROP, the percentage of severe ROP is similar between England (8.26%) and Greece (7.39%). The similar American Academy of Pediatrics [9] criteria used in France and Italy also produced comparably high ROP % rates among preterm infants screened. A range of 25.1–38% was described in Italy and an average of 27.7% in France. Other countries usually follow a set of internal criteria, which are updated at various intervals in accordance with the latest epidemiological studies and the literature on ROP risk factors. Comparing data from the US to data from Europe, both similarities and differences can be observed. In a study by Prakalapakorn et al. [48], a decrease in ROP diagnoses from 37% (2008) to 32% (2018) was observed.

The 2018 result is far from the data presented by countries from various regions of Europe. It is only surpassed by the results from Portugal, in which the percentage of ROP diagnoses in the 2010–2016 period reached 64.1% [18], and 42.3% in 2008–2019 [17], Norway in 2009–2017 (39.6%) [20], and a single center in Florence, Italy (38%) in 2017–2020 [15]. These results are also matched by those of a Swedish study in 2008–2017 (31.9%) and in 2008–2015 (31.9%) [22,24]. 

It is also worth noting that the guidelines of the American Academy of Pediatrics used in the USA, as well as in screening tests in premature infants in European countries, such as France or Italy, lead to similar findings of high percentages of ROP occurrence (France: 27.7%; Italy: 38% and 25.1%; US 2018: 32%) [11,14,15,48].

An interesting aspect is a diversity in the occurrence of ROP with the same screening criteria, and an average similar number of preterm infants screened (Almeida et al. [16], 475 vs. Malheiro et al. [18], 496) in Portugal. One possible reason for the differences in the prevalence of ROP is the different geographic distribution of the three medical centers (140 km–411 km) (Figure 1).

Another explanation may lie in the differences in neonatal care or a higher percentage of extremely preterm infants. It is difficult to make an assessment between the data, both due to the different data collection timespan (2012–2020 vs. 2008–2019 vs. 2010–2016) and some missing data, i.e., average GA and BW, % severe ROP, and percentage of treatment ROP.

Surprisingly, Swedish studies from a similar period (2008–2015 vs. 2007–2018 vs. 2008–2017) showed the same incidence of ROP (31.9%), as well as almost the same percentage of treatment: 5.7% vs. 5.8% vs. 6.1% [22,23,24]. However, these studies were based on the same database (SWEDROP register); therefore, the results may be similar, unlike the Portuguese results, where each occurrence result comes from a different center and, thus, an internal screening database of the clinical center data; hence the differences. By using the generalized database, Sweden can gradually improve its screening guidelines and diagnostic data for ROP therapy based on constant information from the whole region. Similarly to the Sweden research, a unified European database is currently being developed in a project led by researchers from Germany—EU-ROP study, which involves dozens of centers from various European countries, and was recently launched in 2021. Epidemiological data is collected according to a strictly established post-testing scheme contained in detailed questionnaires. Such activities, conducted on a long-term basis, will allow the creation of new, reliable European databases related to the incidence of ROP, as well as to risk factors in ROP, which will perhaps allow for the definition of an appropriate management strategy for new cases of severe ROP [49]. The increase in the diagnosis and treatment rate of ROP in recent years is explained by, among other things, the narrowing of diagnostic criteria in newly issued guidelines (which include the WINROP and G-ROP algorithm) [50,51], a greater number of less developed preterm infants and a lower percentage of live births. In contrast to the increase in the incidence of ROP, the low results of ROP diagnoses and incidence obtained in Swiss studies demonstrate that the decrease in the development of ROP requiring treatment is significantly influenced by the high level of highly specialized neonatal care and the use of modern equipment in NICUs. Therefore, it seems important that more and more European countries join this important multi-center task.

## 5. Conclusions

Our review found that epidemiological data on the incidence of ROP in European countries varies significantly. Currently, it is not possible to standardize screening guidelines due to the notable discrepancies in the tests carried out in different European countries. It is also difficult due to the different health care systems, different guidelines followed in neonatal wards, as well as the genetic variations that cause variations in the course of the disease between premature babies from Eastern and Western Europe, as well as Scandinavian and Balkan countries. Therefore, it is important to spread knowledge about the possibility of joining the EUROP multi-center study conducted by a German center in Europe to standardize our ROP screening procedure.

## Figures and Tables

**Figure 1 jcm-12-03650-f001:**
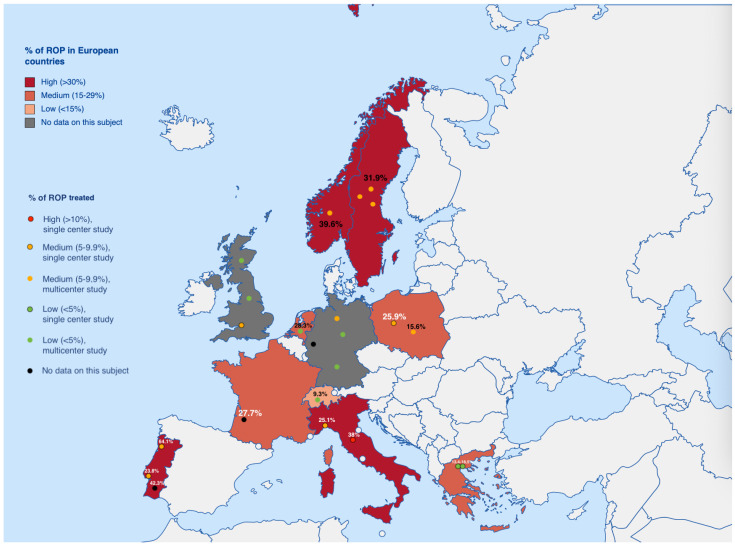
Graphical representation of the % prevalence of ROP in European countries. The color of the text: black, data are from a multi-center study; white, data are from single-center studies. The number of dots indicates the number of surveys from a given country in the period we are describing. If the test was single-center—an attempt was made to put a dot/text in the place where the test was performed. White dots mark countries like: Andorra, Monaco, Malta, Vatican City, San Marino, Liechtenstein.

**Table 1 jcm-12-03650-t001:** Epidemiological data obtained from a review of available findings in the literature from European countries.

	Authors	Years	Database	^a^ ROP Screening (*n*)	ROP Diagnosis (*n*)	^b^ ROP Diagnosis (%)	^b^ Severe ROP	^b^ Treated ROP	Mean BW [g]	Mean GA [Hbd]
England	Wong et al. [26]	2009–2011	NHS operational data National Neonatal Research Database (NNRD) (161 medical units in England)	16,411	No data	No data	No data	3%	No data	No data
	Tavassoli et al. [25]	2009–2015	Bristol Eye Hospital	1152	No data	No data	No data	5.7–13.8%	No data	No data
	Adams et al. [27]	12.2013–12.2014	British Ophthalmic Surveillance Unit and a national collaborative ROP special interest groups (327 group)	8112	No data	No data	8.26%	4%	706	25
France	Chan et al. [11]	2009–2015	Bordeaux Hospital	419	116	27.7%	9.5%	No data	1034 ± 263	28 ± 1.8
Greece	Mataftsi et al. [13]	2004–2015	Papageorgiou General Hospital of Thessaloniki	1178	232	19.7%	7.39%	2.5%	No data	No data
	Moutzouri et al. [12]	2004–2020	Papageorgiou General Hospital of Thessaloniki	1560	288	18.5%	No data	3.14%	776.6 ± 217.6	26 ± 2
The Netherlands	Trzcionkowska et al. [19]	1.01.2010–31.03.2013 (NEDROP)	Neonatal Intensive Care Units of 10 Dutch hospitals	11,295	No data	No data	No data	1.7%	No data	No data
	Trzcionkowska et al. [19]	1.04.2014–31.12.2016 (ETROP)	Neonatal Intensive Care Units of 10 Dutch hospitals		No data	No data	No data		No data	No data
	Trzcionkowska et al. [6]	2017	80 Dutch hospitals	933	264	28.3%	4.4%	No data	No data	No data
Germany	Larsen et al. [28]	2010–2017	IQTIG	52,461	No data	No data	2.9%	2.9%	No data	No data
	Larsen et al. [28]	2012–2016	Bonn/Freiburg		No data	No data	No data	No data	No data	No data
	Larsen et al. [28]	2011–2018	ROP Registry		No data	No data	No data	1.42%	No data	No data
	Akman et al. [29]	2001–2017	Single medical center in Hannover (ROP Registry)	864	No data	No data	No data	7.5%	763 ± 235	25.7 ± 1.8
Norway	Grottenberg et al. [20]	2009–2017	Norwegian Neonatal Network (NNN)	1156	458	39.6%	13.1%	9.5%	830 ± 203	26.2 ± 1.3
Poland	Modrzejewska et al. [31]	2012–2021	GOCC database	97,214	15,190	15.6%	No data	5%	No data	No data
	Chmielarz-Czarnocińska et al. [30]	2016–2019	Adam Mickiewicz Medical University Hospital in Poznan, two large regions of Poland	1772	459	25.9%	No data	6.1%	868 ± 236	26 ± 2
Portugal	Almeida et al. [16]	2012–2020	Department Ophthalmology, Hospital Beatriz Angelo	475	113	23.8%	12.6%	6.1%	No data	No data
	Figueiredo et al. [17]	2008–2019	Department of Ophthalmology and at the Neonatal Unit of the Department of Pediatrics of the Hospital do Espírito Santo de Évora	239	101	42.3%	11.9%	No data	1241.6 ± 310.0	29.8 ± 3.4
	Malheiro et al. [18]	2010–2016	Centro Hospitalar Universitário do Porto	496	275	64.1%	No data	5.8%	1222.4	31.1
Switzerland	Gerull et al. [21]	2006–2015	SwissneoNet registry	6472	No data	9.3%	1.8%	1.2%	No data	No data
Sweden	Holmström et al. [22]	2008–2017	SWEDROP register	8473	2310	31.9%	10.5%	6.1%	No data	No data
	Holmström et al. [24]	2008–2015		5734	1829	31.9%	No data	5.7%	No data	No data
	Pivodic et al. [23]	1.01.2007–31.10.2017		7609	2427	31.9%	No data	5.8%	1119 ± 353	28.1 ± 2.1
Italy	Dani et al. [15]	2017–2020	Careggi University Hospital in Florence	178	67	38%	No data	6.2%	No data	No data
	Caruggi et al. [14]	2015–2020	University of Genoa	475	119	25.1%	No data	5.9%	No data	No data

Abbreviations: *n*, number of premature babies; BW, birth weight; GA, gestational age. ^a^ Number of premature babies excluding missing data and deceased premature babies. ^b^ Ratio of the number of premature babies with the ROP stage described in the column to the number of premature babies screened. Grand Orchestra of Christmas Charity, GOCC—a charitable organization performing, according to its charter, “health care activities involving saving the lives of the ailing, especially children, and working to improve their health, as well as working for health promotion and preventive health care”.

**Table 2 jcm-12-03650-t002:** Screening guidelines of Retinopathy of Prematurity in European countries.

	Authors	Recommendations	Basic Screening Criteria	Extended Screening Criteria
England	Wong et al. [26]	Royal Collage of Paediatrics and Child Health [8]	GA < 32 weeks (up to 30 weeks and 6 days) or BW < 1501 g	GA < 31 weeks (up to 30 weeks and 6 days) or BW < 1251 g
	Tavassoli et al. [25]			
	Adams et al. [27]			
Greece	Mataftsi et al. [13]			
	Moutzouri et al. [12]			
France	Chan et al. [11]	American Academy of Pediatrics [9]	GA ≤ 30 weeks or less (as defined by the attending neonatologist) or BW ≤1500 g	or selected premature infants with a BW between 1500 and 2000 g or a GA of >30 weeks who are believed by their attending pediatrician or neonatologist to be at risk for ROP (such as infants with hypotension requiring inotropic support, infants who received oxygen supplementation for more than a few days, or infants who received oxygen without saturation monitoring)
Italy	Dani et al. [15]			
	Caruggi et al. [14]			
The Netherlands	Trzcionkowska et al. [19] (1 January 2010–31 March 2013)	Old Dutch screening criteria (NEDROP) [32]	GA < 32 weeks and/or BW <1500 g	-
	Trzcionkowska et al. [19] (1 April 2013–2016)	NEDROP-2 [33]	GA < 30 weeks and/or BW < 1250 g and a selection of infants with GA 30–32 weeks	and/or BW 1250–1500 g with at least one of the following risk factors: AV; sepsis, NEC; postnatal glucocorticoids; or cardiotonic drugs
	Trzcionkowska et al. [6]			
Germany	Larsen et al. [28]	German national guideline on ROP screening [34]	GA < 32 weeks, or if GA is unknown, birth weight ≤1500 g	or GA < 37 weeks and > 3 days of oxygen supplementation
	Larsen et al. [28]			
	Larsen et al. [28]			
	Akman et al. [29]	New German guidelines [35,36]	GA < 31 + 0	If there is risk factors, such as over 5 days oxygen administration, ECMO usage, severe NEC, BPD, sepsis, or transfusion-requiring anemia
Norway	Grottenberg et al. [20]	Norwegian ophthalmology society [37]	GA ≤ 32	-
Poland	Modrzejewska et al. [31]	Polish neonatologists and the Section of Pediatric Ophthalmology of the Polish Society of Ophthalmology [38]	GA ≤ 33 with BW ≤ 1800 g	or GA > 33 and BW > 1800 g with cardiovascular respiratory failure, low weight gain and other pathologies associated with prematurity, qualified by neonatologist considering the child’s general condition and high risk of ROP
	Chmielarz-Czarnocińska et al. [30]			
Portugal	Almeida et al. [16]	Portuguese Society of Neonatology [39]	GA ≤ 32 weeks or BW ≤ 1500 g	or BW < 2000 g and prolonged exposure to oxygen, or selected infants who were at higher risk of ROP for having severe disease or who had undergone major surgery (according to the opinion of the attending neonatologist or pediatrician)
	Figueiredo et al. [17]			
	Malheiro et al. [18]			
Switzerland	Gerull et al. [21]	No uniform screening criteria. The criteria was differed slightly between the individual centers.	Mostly included preterm infants GA < 31–32 and/or BW < 1500 g	-
Sweden	Holmström et al. [22]	National guidelines using SWEDROP register [40,41]	GA < 31	-
	Holmström et al. [24]			
	Pivodic et al. [23]			

Abbreviations: BW, birth weight; GA, gestational age; ROP, Retinopathy of Prematurity; AV, artificial ventilation; NEC, necrotizing enterocolitis; ECMO, Extra Corporeal Membrane Oxygenation; BPD, bronchopulmonary dysplasia.

## Data Availability

No new data were created or analyzed in this review. Data sharing does not apply to this article.

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
