# Peer review of "Most Up-to-Date Analysis of Epidemiological Data on the Screening Guidelines and Incidence of Retinopathy of Prematurity in Europe—A Literature Review"

_jcm, 2023, doi:10.3390/jcm12113650_

Round 1
Reviewer 1 Report
General Comments:
The authors take on the challenging task of summarizing the state of ROP in Europe. There is a lot of data to discuss, and each section could be made more uniform to improve the flow.
Including more data about the incidence of prematurity in each population would be of interest to the reader.
Please combine discussion and conclusion sections. Both are not needed and they are repetitive. Suggest to make conclusion section one paragraph maximum.
Specific Comments:
Line 21 This sentence is confusing, “multi-center centers,” please re-write.
Line 22 Please choose one value as the highest incidence for Portugal.
Line 23 Would suggest “national” rather than “internal criteria”
Line 27 The intended use of term “unstructured” here is not clear.
Line 41 PPROM has to be fully defined prior to using abbreviation
Page 7, please replace “cardiotonica”
Line 238, wouldn’t over screening lead to more normal babies and decreased ROP incidence?
Line 244, these percentages do not add close to 100% please correct
Line 252, Why was treatment given to children without ROP?
Overall good quality
Author Response
Author's Reply to the Review Report (Reviewer 1)
The authors of the manuscript thank the reviewers for their detailed review. All comments have been meticulously explained by us and have been added to the text as a supplement. We hope that the current form of the article is still accessible. We are willing to complete the text if you still read it and notice any errors.
Answers for Specific Comments
General Comments:
Including more data about the incidence of prematurity in each population would be of interest to the reader.
Authors: We re-analysed the literature we provided from different countries on the occurrence of ROP. Unfortunately, as we mentioned, the research was carried out in different years, so finding additional literature containing the number of premature babies in a given period in given countries is impossible. Moreover, articles often focus only on the number of preterm infants after screening tests and do not take into account the number of preterm infants born in these centres.
We understand that this data would certainly enrich our article, however, we are unable to include it in it due to unavailability.
Please combine discussion and conclusion sections. Both are not needed and they are repetitive. Suggest to make conclusion section one paragraph maximum.
Authors: The authors agree with the reviewer's valuable comment. Discussion and Conclusions were unified and a short summary was created than before.
Specific Comments:
Line 21 This sentence is confusing, “multi-center centers,” please re-write.
Authors: "Multi-center centers" replaced with "multiple centers" as in the following fragments of the text.
Line 22 Please choose one value as the highest incidence for Portugal.
Authors: Value left: 64.1% , lower value removed: 42.3%
Line 23 Would suggest “national” rather than “internal criteria”
Authors: The authors agree with the reviewer's comment. Replaced "internal" with "national".
Line 27 The intended use of term “unstructured” here is not clear.
Authors: We replaced the word "unstructured" with "varies significantly".
Line 41 PPROM has to be fully defined prior to using abbreviation
Authors: Shortcut explained. Abbreviations for other disease entities have also been added.
Page 7, please replace “cardiotonica”
Authors: Replaced with “Cardiotonic drugs”
Line 238, wouldn’t over screening lead to more normal babies and decreased ROP incidence?
Authors: After carefully analyzing the sentence, the authors agree with the reviewer. Therefore, the sentence „The country screens all premature infants <32 GA, without narrowing down the criteria for ROP risk factors.” is deleted, but we leave the information regarding high mortality and missing data leading to high incidence of ROP.
Line 244, these percentages do not add close to 100% please correct
Authors: Not all premature babies tested and diagnosed with ROP have been treated. The number of preterm infants treated with laser (29.9% (4,536 out of 15,190)) was reported in 2012-2021, and anti-VEGF monotherapy in 2018-2021 (5.6% of preterm infants (301 out of 245, 15,190)), as well as combined - 5.2% of premature infants with 246 ROPs (279 out of 15,190)) due to the lack of prior data on anti-VEGF monotherapy (from 2012-2017). Data should not add up to 100%. (Old line 244 is now 271-276)
Line 252, Why was treatment given to children without ROP?
Authors: The author of this study compared the number of premature infants treated: to the number of premature infants diagnosed with ROP and to the number of premature infants after screening. Numbers in brackets added for clarification. (Old line 252 is now 281-283)
Thank you for the valuable comments in your review. They helped to change some parts of the article, making them clearer and understandable.
Best Regards,
Authors
Monika Modrzejewska & Wiktoria Bosy-GÄ…sior

Reviewer 2 Report
This is an important and well written manuscript regarding a research area , which is of major interest today
Comments. Regarding the quality of grading of ROP in clinic practice depends on which methods you are using.
Binocular or monocular ophthalmoscopy or retcam screening. If possible it could enhance the quality of the manuscript, if at least some of the publications has mentioned that.
It could be, that retcam examination earlier detect ROP and that could be an effect for incidence of this disorder in some countries, but I understand that this data could be difficult to get.
Author Response
Author's Reply to the Review Report (Reviewer 2)
The authors of the manuscript thank the reviewers for their detailed review. All comments have been meticulously explained by us and have been added to the text as a supplement. We hope that the current form of the article is still accessible. We are willing to complete the text if you still read it and notice any errors.
Answers for Specific Comments
Reviewer 2
Comments. Regarding the quality of grading of ROP in clinic practice depends on which methods you are using.
Binocular or monocular ophthalmoscopy or retcam screening. If possible it could enhance the quality of the manuscript, if at least some of the publications has mentioned that.
It could be, that retcam examination earlier detect ROP and that could be an effect for incidence of this disorder in some countries, but I understand that this data could be difficult to get.
Authors: We checked in each of the publications we listed whether it was mentioned how thoroughly the screening of premature babies was carried out in each country. Unfortunately, none of them contain such information, so as the reviewer mentioned - it would be difficult for us to obtain information about the use of the RetCam device.
In accordance with the reviewer's comment, we added a fragment about the general use of RetCam in ROP screening, as well as during further ophthalmological visits to check the condition of the retina after the introduction of treatment.
In addition to the fragment about the RetCam device, we have also added additional information about the new screening method, which involves the use of a smartphone. After a few refinements, it may turn out to be a very future-proof method.
Thank you for the valuable comments in your review. They helped to change some parts of the article, making them clearer and understandable.
Best Regards,
Authors
Monika Modrzejewska & Wiktoria Bosy-GÄ…sior

Round 2
Reviewer 1 Report
The authors provide a a revised manuscript which is appreciated.
Line 276, the percentages here are still misleading since most other sections list % of treatment modality compared to all treated babies not all ROP (ie it should add up to 100%). See Portugal paragraph for example of correct description. All sections need to have uniform statistics or the paper is challenging to read.
Author Response
Author's Reply to the Review Report (Reviewer 1)
The authors would like to thank you for your positive review and appreciation of our previous fixes.
Answers for Specific Comments
General Comments:
Line 276, the percentages here are still misleading since most other sections list % of treatment modality compared to all treated babies not all ROP (ie it should add up to 100%). See Portugal paragraph for example of correct description. All sections need to have uniform statistics or the paper is challenging to read.
Authors: As requested by the reviewer, we corrected the section on treatment in Poland so that the sum equals 100%.
Thank you for the valuable comments in your review.
Best Regards,
Authors
Monika Modrzejewska & Wiktoria Bosy-GÄ…sior